# Inadequate preparedness for response to COVID-19 is associated with stress and burnout among healthcare workers in Ghana

Patience A. Afulani[1,2], Akua O. Gyamerah[3¤]*, Jerry J. Nutor[4], Amos Laar[5], Raymond A. Aborigo[6], Hawa Malechi[7], Mona Sterling[2], John K. Awoonor-Williams[8]

1 Department of Epidemiology & Biostatistics, University of California, San Francisco, San Francisco, California, United States of America, 2 Institute for Global Health Sciences, University of California, San Francisco, San Francisco, California, United States of America, 3 Center for AIDS Prevention Studies, University of California, San Francisco, San Francisco, California, United States of America, 4 Department of Family Health Care Nursing, University of California, San Francisco, San Francisco, California, United States of America, 5 Department of Population, Family and Reproductive Health, School of Public Health, University of Ghana, Accra, Ghana, 6 Navrongo Health Research Centre, Navrongo, Ghana, 7 Tamale Teaching Hospital, Tamale, Ghana, 8 Policy Planning Monitoring and Evaluation Division, Ghana Health Service, Accra, Ghana

¤ Current address: Department of Community Health Systems, University of California, San Francisco, San Francisco, California, United States of America
* akua.gyamerah@ucsf.edu

**Data Availability Statement:** All relevant data are within the paper and its Supporting information files.

## Abstract

### Introduction

The COVID-19 pandemic has compounded the global crisis of stress and burnout among healthcare workers. But few studies have empirically examined the factors driving these outcomes in Africa. Our study examined associations between perceived preparedness to respond to the COVID-19 pandemic and healthcare worker stress and burnout and identified potential mediating factors among healthcare workers in Ghana.

### Methods

Healthcare workers in Ghana completed a cross-sectional self-administered online survey from April to May 2020; 414 and 409 completed stress and burnout questions, respectively. Perceived preparedness, stress, and burnout were measured using validated psychosocial scales. We assessed associations using linear regressions with robust standard errors.

### Results

The average score for preparedness was 24 (SD = 8.8), 16.3 (SD = 5.9) for stress, and 37.4 (SD = 15.5) for burnout. In multivariate analysis, healthcare workers who felt somewhat prepared and prepared had lower stress (β = -1.89, 95% CI: -3.49 to -0.30 and β = -2.66, 95% CI: -4.48 to -0.84) and burnout (β = -7.74, 95% CI: -11.8 to -3.64 and β = -9.25, 95% CI: -14.1 to −4.41) scores than those who did not feel prepared. Appreciation from management and family support were associated with lower stress and burnout, while fear of infection was associated with higher stress and burnout. Fear of infection partially mediated the

**Funding:** This research is financially supported by the University of California, San Francisco COVID-19 Related Rapid Research Pilot Initiative. The funders had no role in study design, data collection and analysis, decision to publish, or preparation of the manuscript.

**Competing interests:** The authors have declared that no competing interests exist.

relationship between perceived preparedness and stress/burnout, accounting for about 16 to 17% of the effect.

## Conclusions

Low perceived preparedness to respond to COVID-19 increases stress and burnout, and this is partly through fear of infection. Interventions, incentives, and health systemic changes to increase healthcare workers' morale and capacity to respond to the pandemic are needed.

## Introduction

The novel coronavirus disease (COVID-19) has become a major health crisis of our generation. The pandemic had affected over 19.8 million people and claimed the lives of over 733,000 people as of August 10th, 2020 [1]. According to the World Health Organization (WHO), Healthcare workers (HCWs) in particular have been disproportionately impacted by COVID-19, accounting for over 10% of global infections [2]. In Africa, over 10,000 HCWs across 40 countries contracted COVID-19 as of July 23rd, 2020 [3]. Underlying the epidemic among these frontline workers are various factors that may be shaping HCWs' risk of COVID-19, including preparedness indicators such as inadequate training, protocols, knowledge, personal protection equipment (PPE), as well as weak health systems, slow national responses, and poor political leadership [2, 4, 5]. Yet, emerging data indicate that providers across the globe are inadequately prepared to respond to the pandemic [5, 6].

The scale and rapid spread of COVID-19, combined with inadequate preparedness, may be contributing to HCW stress and burnout—two psychological indicators that reached crisis levels among HCWs globally prior to COVID-19 [7, 8]. Chronic work-related stress, when not adequately managed, leads to burnout, which manifests as physical, cognitive, and emotional exhaustion and depersonalization (feelings of negativism, cynicism, or detachment from one's job), and reduced professional efficacy [9]. Burnout leads to lower productivity and effectiveness, decreased job satisfaction and commitment, and poor quality care, with risks to patient safety [10, 11]. Stress and burnout is also associated with poor health outcomes such as depression, cardiovascular disease, and premature mortality [12, 13]. Moreover, HCW burnout is expensive for the health system given its associations with care quality, absenteeism, and workforce turnover, and is, thus, critical to examine [14, 15].

Since the WHO declared COVID-19 a global pandemic, a growing number of studies have examined its psychological impact on frontline workers [16–19]. A qualitative study among HCWs in China found that challenges experienced in responding to COVID-19 included exhaustion from prolonged use of protective gear and heavy workloads, and fear of infection and infecting others, while social support and self-management strategies helped HCWs cope with distress [20]. Additionally, a systematic review found that HCWs are experiencing psychological distresses due to COVID-19, with the following pooled estimates for anxiety (26%), depression (25%), distress (35%), stress (40%), insomnia (32%), and PTSD (3% to 16%) [19]. None of these studies were in Africa.

Inadequate preparedness has been linked to various psychological outcomes among HCWs in prior epidemics outbreaks [21]. But few studies have empirically examined this in the context of COVID-19 and no studies, to our knowledge, have specifically examined the psychological impact of *perceived preparedness* among HCWs to respond to the COVID-19. A

nationwide survey examining psychological distress among the general population in China during the COVID-19 epidemic found that preparedness indicators (e.g., having effective prevention and control measures and a highly efficient health system) were protective against psychological distress [22]. In Africa, where health systems are constrained and underfunded [23], no empirical studies in the context of COVID-19 have reported on this issue to date for HCWs. However, a prior systematic review found that, generally, burnout is high among physicians and even higher among nurses in Africa and that drivers of burnout include lack of social support, long work hours and understaffing, and professional and interpersonal conflicts [24, 25].

Due to shortage of staff and limited resources, HCWs in African settings have been working under excessive workloads and psychologically charged environments where demand outweighs capacity [26, 27]. In Ghana, which has the third highest number of COVID-19 cases in Africa and over 2,000 HCWs infected [28, 29], stress and burnout may be even higher among HCWs. Previous studies assessing HCWs' preparedness for the Ebola outbreak in Ghana found that providers felt inadequately prepared to respond, and reported issues such as inadequate PPE and staff [30–33]. Our study contributes to addressing the gap in the literature on the psychological impact of COVID-19 on African HCWs by examining HCW stress and burnout and associations with perceived preparedness to respond to COVID-19 and other factors in Ghana.

## Materials and methods

### Study setting

Ghana recorded its first two cases of COVID-19 on March 12[th], 2020. Since then, the epidemic in Ghana has grown exponentially, with 41,212 cases and 215 deaths as of August 10[th], 2020, making it the country with the third highest number of cases in Africa and 51[st] globally [1, 29]. Ghana has a constrained health system, with a population of approximately 30 million, an estimated 1.8 medical doctors and 42 nurses and midwives per 10,000 population, and less than one hospital bed per 1,000 people [23, 34–36]. The increasing number of cases within an overburdened healthcare infrastructure is, therefore, a major source of concern for many HCWs. In addition, HCWs have expressed fear of coronavirus infection due to concerns about inadequate PPE and testing, sparking threats of industrial strike actions by nurses and doctors in Ghana [37, 38]. New data on the high number of COVID-19 cases among HCWs, including six deaths, has elevated this fear and, raised renewed concerns about the potential catastrophic effects of a weak health system and lack of HCW preparedness [39, 40].

### Study design

This is a cross-sectional study conducted with HCWs in Ghana (i.e., nurses, physicians, and allied health workers) from April 17[th], 2020 to May 31[st], 2020. We used a convenience sampling approach to recruit HCWs virtually through advertising on diverse online and social media platforms (WhatsApp, Facebook, and direct messaging), and invited them to complete a self-administered online survey through a link in the ad. Eligibility criteria was identifying as a HCW based in Ghana. To maximize representativeness in our sample, we disseminated survey links to Facebook and WhatsApp pages of different professional groups, graduation year groups, and regional groups of HCWs, as well as to leaders of professional organizations and Ghana Health Service directors to share with members of their groups. No incentives were provided, and respondents had the option of skipping questions. The survey was conducted in English and included questions on demographics, perceived preparedness, stress, burnout, and other questions relevant to the pandemic response. The survey was pretested with 10

HCWs in Ghana by sending them the link to complete the survey and provide feedback. Feedback from the pretest was used to finalize the survey. Providers consented to the study by completing the survey. A total of 646 HCWs started the survey (i.e., answered the first question in the survey). Additional study methods can be found in a prior manuscript on HCWs' perceived preparedness to respond to COVID-19 [41].

## Measures

**Dependent variables: Stress and burnout.** The two outcome variables—stress and burnout—were measured using validated psychosocial measures. Stress was assessed using the 10-item Cohen perceived stress scale, which captures people's feelings and thoughts in the past month [42]. Questions relate to how nervous or stressed, unpredictable, uncontrollable, and overloaded respondents find their lives (S1 Appendix). Each question is on a scale of 0 (never) to 4 (very often). Burnout was assessed using the 14-item Shirom-Melamed Burnout measure (SMBM), which assesses feelings at work in the past month [43]. Questions capture three domains of burnout: physical fatigue, emotional exhaustion, and cognitive weariness, with responses options ranging from 1 (never or almost never) to 7 (always or almost always) (S2 Appendix).

**Independent variables.** The key predictor in this analysis is *perceived preparedness to respond to COVID-19*, which was assessed using a 15-item scale developed by our team. The questions capture personal, facility, and psychological preparedness for prevention, diagnoses, management, and education regarding COVID-19. Each question has response options from 0 (not prepared at all) to 3 (very prepared), with options for "I don't know about this (4), and "Not applicable to my role" (5) (S3 Appendix). The scale development process is described elsewhere [41].

Other independent variables included *feeling of appreciation*, *support*, and *communication from management*; *family support*, *ability to isolate at home without exposing family*, *fear of contracting COVID-19*, *confidence in being cared for if infected*, *COVID-19 training*; *availability of PPE*, *isolation ward*, and *protocols for COVID-19*; *perceived knowledge* of how to manage COVID-19 (S4 Appendix), and provider and facility characteristics.

## Analysis

We used data from respondents who answered all questions on stress and burnout and relevant predictors for this analysis. Many respondents (n = 216) did not get to the stress and burnout questions, which were among the final set of questions, because they ended the survey prematurely. We, therefore, excluded these respondents, as well as an additional 16 and 21 respondents who started but did not complete the questions on stress and burnout, respectively. The resulting analytic samples, which are overlapping, are 414 and 409 for the stress and burnout, respectively.

We examined the distribution of variables using descriptive statistics and created summative scores for stress, burnout, and preparedness. Factor analysis showed all three scales had good construct validity with all items in each scale loading on one dominant factor with eigenvalues greater than three. The scales also had good internal consistency with Cronbach alpha of 0.79 for stress, 0.94 for burnout, and 0.91 for preparedness. Before creating summative scores, items were recoded such that higher scores indicate higher stress, burnout, and preparedness. For the preparedness score, we coded response options to range from 0 to 3 by recoding 4 (I don't know about this) to 0 (not at all prepared) and 5 (not applicable to my role) to 2 (prepared). Stress scores range from 0 to 40. Scores of 0–13 are considered low stress, 14–26 moderate stress, and 27–40 high stress [42]. Burnout scores range from 14–98—rescaled to

1–7 by dividing by total number of items for ease of comparison with sub-domains. Scores of ≤2.0 are considered no burnout, 2–3.74 moderate burnout, and ≥3.75 as high burnout [44]. We used the same cutoffs for burnout domains. Preparedness scores range from 0–45. We categorized scores less than 15 as "not at all prepared"; scores 15 to 29 as "somewhat prepared," and ≥30 as "prepared" [41].

We used the continuous scores for the outcomes in linear regressions with robust standard errors to examine the associations with various predictors. The burnout score was slightly skewed to the right, which was corrected with a log transformation. For ease of interpretation, we used the untransformed variable for the main analysis and conducted sensitivity analysis with the log transformed variable. We built multivariate models by gradually adding demographic and other independent variables that were significant in the bivariate analysis and testing for model fit and collinearity. Finally, we examined if the relationships between perceived preparedness and both stress and burnout were mediated by fear of infection using the difference of coefficients (c-c') method. The mediated or indirect effect is the difference in the coefficients in the model without the mediator (total effect: c) and that in the model with the mediator (direct effect: c'). The proportion mediated is ((c-c')/c) [45, 46]. We also examined if the associations were moderated by type of health provider, appreciation from management, and family support. In additional analysis, we ran the models with preparedness as a continuous variable and with the outcomes as binary variables.

## Ethical approval

Ethical approval was obtained from the University of California, San Francisco (#20–30656) and the Navrongo Health Research Centre (#NHRCIRB374).

## Results

### Descriptive results

About 20% were doctors, 62% nurses (including midwives and medical/physician assistants) and 18% other professionals, including medical laboratory professionals, disease control officers, nutritionists and other allied health care workers (Table 1). About 26% worked in teaching hospitals, 59% in other public hospitals (e.g., regional and district hospitals and health centers), and 15% in private facilities. Approximately 23% work in the Greater Accra and Ashanti regions (the initial epicenters), 23% from Northern region, and the rest from other regions. There were at least 10 respondents from each the 16 regions of the country, except for the Bono and Ahafo regions, which had less than five respondents. The average age of respondents was 34.2 years (SD = 6.0), with 8.2 years of professional experience (SD = 5.6). About half were female.

The average stress score was 16.3 (SD = 5.9), with 64% having moderate stress and 4% high stress. Average burnout score was 37.4 (SD = 15.5), with 47% having low burnout and 20% high burnout. About 33%, 15%, and 23%, respectively, had high values for physical exhaustion, emotional exhaustion, and cognitive weariness. Average preparedness score was 24 (SD = 8.8), with 56.9% somewhat prepared and 27.5% prepared (Table 1). About 44% perceived management was appreciative or very appreciative of their efforts and 55% perceived communication from management was good or very good. Additionally, 46% were fearful or very fearful of contracting COVID-19 and only 20% were confident or very confident that they would be adequately cared for in their facility if they got infected. About 67% felt their families were supportive or very supportive of their work, and 33% were certain of a place to isolate at home without exposing their family if they were infected. Distribution of other variables shown in Table 1.

**Table 1. Participant demographics and univariate distribution of study variables, healthcare workers in Ghana.**

| Variables | Stress sample (N = 414) | | Burnout sample (N = 409) | |
|---|---|---|---|---|
| | No. | % | No. | % |
| Provider type | | | | |
| Doctor | 82 | 19.8 | 81 | 19.8 |
| Nurse/related | 259 | 62.6 | 256 | 62.6 |
| Other [a] | 73 | 17.6 | 72 | 17.6 |
| Facility type | | | | |
| Teaching hospital | 109 | 26.3 | 108 | 26.4 |
| Regional/district hospital | 119 | 28.7 | 117 | 28.6 |
| Health center/Other govt facility | 125 | 30.2 | 124 | 30.3 |
| Private/mission facility | 61 | 14.7 | 60 | 14.7 |
| Region | | | | |
| Greater Accra/Ashanti | 94 | 22.7 | 93 | 22.7 |
| Northern region | 94 | 22.7 | 92 | 22.5 |
| Other Northern | 96 | 23.2 | 95 | 23.2 |
| Other Southern | 130 | 31.4 | 129 | 31.5 |
| Years of experience | | | | |
| 5 or less years | 134 | 32.4 | 133 | 32.5 |
| 6 to 10 years | 173 | 41.8 | 169 | 41.3 |
| More than 10 years | 107 | 25.8 | 107 | 26.2 |
| Ages | | | | |
| Less than 30 | 113 | 27.5 | 111 | 27.3 |
| 30 to 39 | 234 | 56.9 | 232 | 57.1 |
| 40 to 73 | 64 | 15.6 | 63 | 15.5 |
| Gender | | | | |
| Male | 210 | 50.7 | 208 | 50.9 |
| Female | 204 | 49.3 | 201 | 49.1 |
| No. of children | | | | |
| No children | 124 | 30.6 | 121 | 30.2 |
| 1 or 2 children | 189 | 46.7 | 189 | 47.2 |
| 3 to 6 children | 92 | 22.7 | 90 | 22.5 |
| Marital status [b] | | | | |
| Single | 120 | 29 | 119 | 29.1 |
| Married | 294 | 71 | 290 | 70.9 |
| Perceived stress | | | | |
| Low stress | 130 | 31.4 | | |
| Moderate stress | 266 | 64.3 | | |
| High stress | 18 | 4.3 | | |
| Burnout | | | | |
| No burnout | | | 135 | 33.0 |
| Low burnout | | | 192 | 46.9 |
| High burnout | | | 82 | 20.0 |
| Physical fatigue | | | | |
| No fatigue | | | 110 | 26.9 |
| Low fatigue | | | 163 | 39.9 |
| High fatigue | | | 136 | 33.3 |
| Emotional exhaustion | | | | |

(*Continued*)

**Table 1.** (Continued)

| Variables | Stress sample (N = 414) | | Burnout sample (N = 409) | |
|---|---|---|---|---|
| | No. | % | No. | % |
| No exhaustion | | | 251 | 61.4 |
| Low exhaustion | | | 97 | 23.7 |
| High exhaustion | | | 61 | 14.9 |
| Cognitive weariness | | | | |
| No weariness | | | 196 | 47.9 |
| Low weariness | | | 118 | 28.9 |
| High weariness | | | 95 | 23.2 |
| Preparedness | | | | |
| Not at all prepared | 65 | 15.7 | 63 | 15.4 |
| Somewhat prepared | 235 | 56.8 | 233 | 57.0 |
| Prepared | 114 | 27.5 | 113 | 27.6 |
| Appreciation from management | | | | |
| Not at all appreciative | 61 | 14.7 | 58 | 14.2 |
| Somewhat appreciative | 173 | 41.8 | 172 | 42.1 |
| Appreciative | 146 | 35.3 | 145 | 35.5 |
| Very appreciative | 34 | 8.2 | 34 | 8.3 |
| Support from management | | | | |
| Not at all supportive | 51 | 12.3 | 49 | 12.0 |
| A little supportive | 218 | 52.7 | 215 | 52.6 |
| Supportive | 123 | 29.7 | 123 | 30.1 |
| Very supportive | 22 | 5.3 | 22 | 5.4 |
| Communication from management | | | | |
| Very poor communication | 49 | 11.9 | 47 | 11.5 |
| Poor communication | 136 | 32.9 | 137 | 33.6 |
| Good communication | 192 | 46.5 | 188 | 46.1 |
| Very good communication | 36 | 8.7 | 36 | 8.8 |
| Fearful of contracting COVID-19 | | | | |
| Not fearful | 53 | 12.8 | 52 | 12.7 |
| A little fearful | 170 | 41.1 | 169 | 41.3 |
| Fearful | 102 | 24.6 | 102 | 24.9 |
| Very fearful | 89 | 21.5 | 86 | 21.0 |
| Confidence in being cared for if infected | | | | |
| Not confident | 181 | 43.7 | 178 | 43.5 |
| A little confident | 151 | 36.5 | 151 | 36.9 |
| Confident | 70 | 16.9 | 68 | 16.6 |
| Very confident | 12 | 2.9 | 12 | 2.9 |
| Support from family | | | | |
| Not at all supportive | 24 | 5.8 | 23 | 5.6 |
| A little supportive | 111 | 26.8 | 108 | 26.4 |
| Supportive | 182 | 44 | 181 | 44.3 |
| Very supportive | 97 | 23.4 | 97 | 23.7 |
| Ability to isolate at home if infected | | | | |
| No | 229 | 55.3 | 226 | 55.3 |
| Somewhat | 51 | 12.3 | 52 | 12.7 |
| Yes | 134 | 32.4 | 131 | 32.0 |

(*Continued*)

**Table 1.** (Continued)

| Variables | Stress sample (N = 414) | | Burnout sample (N = 409) | |
|---|---|---|---|---|
| | No. | % | No. | % |
| Training on COVID-19 | | | | |
| No | 187 | 45.2 | 184 | 45 |
| Yes | 227 | 54.8 | 225 | 55 |
| Facility has adequate PPEs | | | | |
| No | 312 | 75.4 | 308 | 75.3 |
| Yes | 28 | 6.8 | 27 | 6.6 |
| I don't know | 74 | 17.9 | 74 | 18.1 |
| Facility has COVID-19 isolation ward | | | | |
| No | 125 | 30.3 | 124 | 30.4 |
| Yes | 275 | 66.6 | 271 | 66.4 |
| I don't know | 13 | 3.1 | 13 | 3.2 |
| Facility has protocol for screening for COVID-19 | | | | |
| No | 66 | 15.9 | 65 | 15.9 |
| Yes | 333 | 80.4 | 330 | 80.7 |
| I don't know | 15 | 3.6 | 14 | 3.4 |
| Facility has protocol for managing COVID-19 | | | | |
| No | 147 | 35.5 | 145 | 35.5 |
| Yes | 202 | 48.8 | 200 | 48.9 |
| I don't know | 65 | 15.7 | 64 | 15.6 |
| Guidelines to report suspected COVID-19 | | | | |
| No | 78 | 18.8 | 76 | 18.6 |
| Yes | 318 | 76.8 | 315 | 77 |
| I don't know | 18 | 4.3 | 18 | 4.4 |
| Know what to do if COVID-19 suspected | | | | |
| No | 21 | 5.1 | 21 | 5.1 |
| Somewhat | 118 | 28.5 | 118 | 28.9 |
| Yes | 275 | 66.4 | 270 | 66 |
| Know how to manage a confirmed case of COVID-19 | | | | |
| No | 145 | 35.1 | 142 | 34.8 |
| Somewhat | 137 | 33.2 | 137 | 33.6 |
| Yes | 87 | 21.1 | 86 | 21.1 |
| Not applicable to my role | 44 | 10.7 | 43 | 10.5 |

Notes:

[a] This includes other health care professionals such as medical laboratory professionals, disease control officers, nutritionists, and other allied health care workers;

[b] The married category includes 10 people (2%) who were previously married (widowed, separated, or divorced).

## Bivariate results

In the bivariate analysis (Table 2), higher perceived preparedness was associated with lower perceived stress and burnout. The average stress and burnout scores among those who felt prepared was 14 (SD = 5.1) and 33 (SD = 13.5), respectively, compared to 19 (SD = 6.1) and 47 (SD = 15.3), respectively, for those who did not feel at all prepared. Burnout scores among other HCWs were lower than that of doctors and nurses. HCWs in Northern region had lower stress than those in Greater Accra and Ashanti regions and HCWs in other southern regions

**Table 2. Bivariate distributions of stress and burnout among healthcare workers in Ghana by independent variables.**

| | Stress Scores (N = 414) | | | | | Burnout scores (N = 409) | | | | |
|---|---|---|---|---|---|---|---|---|---|---|
| | N | Mean | Sd | β | [95% CI] | N | Mean | Sd | β | [95% CI] |
| **Total** | 414 | 16.3 | 5.9 | | | 409 | 37.4 | 15.5 | | |
| Preparedness | | | | | | | | | | |
| Not at all prepared | 65 | 19.2 | 6.1 | 0 | [0 0] | 63 | 46.9 | 15.3 | 0 | [0 0] |
| A little prepared | 235 | 16.4 | 5.8 | -2.82*** | [-4.38 -1.26] | 233 | 37.2 | 15.3 | -9.13*** | [-13.2 -5.07] |
| Prepared | 114 | 14.4 | 5.1 | -4.84*** | [-6.58 -3.11] | 113 | 32.5 | 13.5 | -14.0*** | [-18.5 -9.47] |
| Provider type | | | | | | | | | | |
| Doctor | 82 | 16.0 | 6.4 | 0 | [0 0] | 81 | 39.9 | 16.8 | 0 | [0 0] |
| Nurse/related | 259 | 16.4 | 5.7 | 0.43 | [-1.03 1.89] | 256 | 37.4 | 15.1 | -2.03 | [-5.88 1.82] |
| Other | 73 | 16.3 | 5.7 | 0.43 | [-1.42 2.28] | 72 | 34.4 | 14.9 | -5.37* | [-10.3 -0.49] |
| Facility type | | | | | | | | | | |
| Teaching hospital | 109 | 15.5 | 5.6 | 0 | [0 0] | 108 | 38.3 | 14.6 | 0 | [0 0] |
| Regional/district hospital | 119 | 16.7 | 5.8 | 1.2 | [-0.33 2.72] | 117 | 36.8 | 14.9 | -1.05 | [-5.13 3.03] |
| Health center/Other govt facility | 125 | 16.5 | 6.1 | 1.1 | [-0.41 2.60] | 124 | 36.7 | 15.7 | -1.11 | [-5.11 2.89] |
| Private/mission facility | 61 | 16.5 | 6.0 | 1.04 | [-0.80 2.88] | 60 | 38.2 | 17.7 | 0.32 | [-4.59 5.23] |
| Region | | | | | | | | | | |
| Greater Accra/Ashanti | 94 | 17.1 | 5.8 | 0 | [0 0] | 93 | 40.1 | 15.3 | 0 | [0 0] |
| Northern region | 94 | 14.6 | 6.0 | -2.46** | [-4.11 -0.80] | 92 | 36.8 | 15.0 | -3.69 | [-8.17 0.79] |
| Other Northern | 96 | 17.6 | 5.8 | 0.52 | [-1.13 2.17] | 95 | 38.2 | 17.1 | -1.9 | [-6.32 2.52] |
| Other Southern | 130 | 16.0 | 5.6 | -1.05 | [-2.58 0.49] | 129 | 35.3 | 14.4 | -4.36* | [-8.48 -0.24] |
| Years of experience | | | | | | | | | | |
| 5 or less years | 134 | 15.7 | 5.8 | 0 | [0 0] | 133 | 36.4 | 15.5 | 0 | [0 0] |
| 6 to 10 years | 173 | 17.2 | 6.1 | 1.46* | [0.14 2.77] | 169 | 38.5 | 15.7 | 2.72 | [-0.79 6.23] |
| More than 10 years | 107 | 15.6 | 5.4 | -0.018 | [-1.50 1.46] | 107 | 36.7 | 15.1 | 0.32 | [-3.63 4.26] |
| Ages | | | | | | | | | | |
| Less than 30 | 113 | 15.7 | 5.7 | 0 | [0 0] | 111 | 37.7 | 15.1 | 0 | [0 0] |
| 30 to 39 | 234 | 16.8 | 5.9 | 1.06 | [-0.25 2.37] | 232 | 37.2 | 16.0 | -0.43 | [-3.94 3.08] |
| 40 to 73 | 64 | 15.4 | 5.7 | -0.3 | [-2.09 1.50] | 63 | 37.1 | 14.4 | -0.78 | [-5.55 3.99] |
| Gender | | | | | | | | | | |
| Male | 210 | 15.5 | 5.6 | 0 | [0 0] | 208 | 34.4 | 15.4 | 0 | [0 0] |
| Female | 204 | 17.1 | 6.0 | 1.52** | [0.40 2.64] | 201 | 40.4 | 15.0 | 6.25*** | [3.31 9.19] |
| No. of children | | | | | | | | | | |
| No children | 124 | 16.0 | 5.5 | 0 | [0 0] | 121 | 37.9 | 15.9 | 0 | [0 0] |
| 1 or 2 children | 189 | 17.0 | 5.8 | 1.02 | [-0.31 2.36] | 189 | 37.5 | 15.4 | -0.53 | [-4.09 3.04] |
| 3 to 6 children | 92 | 15.4 | 6.4 | -0.57 | [-2.16 1.01] | 90 | 36.3 | 15.6 | -1.67 | [-5.92 2.58] |
| Marital status | | | | | | | | | | |
| Single | 120 | 16.4 | 5.3 | 0 | [0 0] | 119 | 39.0 | 15.8 | 0 | [0 0] |
| Married | 294 | 16.2 | 6.1 | -0.13 | [-1.38 1.11] | 290 | 36.7 | 15.3 | -2.05 | [-5.34 1.24] |
| Appreciation from management | | | | | | | | | | |
| Not at all appreciative | 61 | 18.3 | 6.2 | 0 | [0 0] | 58 | 46.7 | 19.1 | 0 | [0 0] |
| Somewhat appreciative | 173 | 17.2 | 5.6 | -1.06 | [-2.72 0.60] | 172 | 39.1 | 14.4 | -7.41** | [-11.8 -3.01] |
| Appreciative | 146 | 15.0 | 5.5 | -3.23*** | [-4.93 -1.53] | 145 | 33.7 | 13.5 | -12.8*** | [-17.3 -8.36] |
| Very appreciative | 34 | 13.1 | 5.6 | -5.22*** | [-7.61 -2.84] | 34 | 28.7 | 12.6 | -18.0*** | [-24.3 -11.7] |
| Support from management | | | | | | | | | | |
| Not at all supportive | 51 | 18.1 | 7.1 | 0 | [0 0] | 49 | 45.6 | 18.8 | 0 | [0 0] |
| A little supportive | 218 | 16.5 | 5.5 | -1.57 | [-3.35 0.20] | 215 | 37.7 | 14.8 | -7.41** | [-12.1 -2.72] |
| Supportive | 123 | 15.5 | 5.7 | -2.61** | [-4.51 -0.71] | 123 | 34.2 | 14.0 | -10.9*** | [-15.9 -5.89] |

(*Continued*)

**Table 2.** (Continued)

| | Stress Scores (N = 414) | | | | Burnout scores (N = 409) | | | |
|---|---|---|---|---|---|---|---|---|
| | N | Mean | Sd | β | [95% CI] | N | Mean | Sd | β | [95% CI] |
| Very supportive | 22 | 14.0 | 6.1 | -4.07** | [-6.98 -1.16] | 22 | 33.8 | 16.5 | -11.4** | [-19.1 -3.78] |
| Communication from management | | | | | | | | | | |
| Very poor communication | 49 | 19.3 | 6.7 | 0 | [0 0] | 47 | 45.8 | 15.8 | 0 | [0 0] |
| Poor communication | 136 | 16.9 | 5.5 | -2.34* | [-4.21 -0.46] | 137 | 39.9 | 15.5 | -5.97* | [-11.0 -0.96] |
| Good communication | 192 | 15.3 | 5.6 | -3.97*** | [-5.77 -2.17] | 188 | 34.6 | 14.8 | -11.3*** | [-16.1 -6.45] |
| Very good communication | 36 | 15.0 | 5.9 | -4.29*** | [-6.76 -1.81] | 36 | 31.6 | 12.8 | -13.1*** | [-19.6 -6.61] |
| Fearful of contracting COVID-19 | | | | | | | | | | |
| Not fearful | 53 | 13.5 | 5.7 | 0 | [0 0] | 52 | 30.2 | 14.4 | 0 | [0 0] |
| A little fearful | 170 | 15.6 | 5.6 | 2.22* | [0.46 3.98] | 169 | 35.6 | 13.6 | 5.32* | [0.69 9.95] |
| Fearful | 102 | 17.1 | 5.4 | 3.69*** | [1.80 5.59] | 102 | 38.5 | 15.5 | 8.32** | [3.33 13.3] |
| Very fearful | 89 | 18.2 | 6.1 | 4.76*** | [2.82 6.70] | 86 | 43.8 | 17.1 | 13.5*** | [8.36 18.6] |
| Confidence in being cared for if infected | | | | | | | | | | |
| Not confident | 181 | 17.4 | 6.3 | 0 | [0 0] | 178 | 40.9 | 16.7 | 0 | [0 0] |
| A little confident | 151 | 15.9 | 5.3 | -1.48* | [-2.72 -0.23] | 151 | 35.5 | 13.6 | -4.66** | [-7.93 -1.39] |
| Confident | 70 | 15.0 | 5.4 | -2.38** | [-3.98 -0.79] | 68 | 33.6 | 14.2 | -7.50*** | [-11.7 -3.29] |
| Very confident | 12 | 12.8 | 5.4 | -4.54** | [-7.92 -1.15] | 12 | 30.7 | 16.6 | -10.1* | [-19.0 -1.12] |
| Support from family | | | | | | | | | | |
| Not at all supportive | 24 | 23.5 | 5.9 | 0 | [0 0] | 23 | 54.9 | 17.4 | 0 | [0 0] |
| A little supportive | 111 | 16.9 | 5.9 | -6.62*** | [-9.08 -4.17] | 108 | 39.1 | 15.4 | -15.2*** | [-21.9 -8.53] |
| Supportive | 182 | 15.6 | 5.2 | -7.98*** | [-10.3 -5.61] | 181 | 35.2 | 14.3 | -19.4*** | [-25.9 -13.0] |
| Very supportive | 97 | 15.2 | 5.8 | -8.39*** | [-10.9 -5.90] | 97 | 35.4 | 14.5 | -19.9*** | [-26.6 -13.1] |
| Ability to isolate at home if infected | | | | | | | | | | |
| No | 229 | 16.4 | 6.0 | 0 | [0 0] | 226 | 37.9 | 15.7 | 0 | [0 0] |
| Somewhat | 51 | 16.7 | 5.0 | 0.27 | [-1.52 2.05] | 52 | 38.9 | 14.7 | 0.46 | [-4.23 5.14] |
| Yes | 134 | 15.9 | 5.9 | -0.53 | [-1.78 0.72] | 131 | 35.9 | 15.3 | -2.07 | [-5.41 1.26] |
| Training on COVID-19 | | | | | | | | | | |
| No | 187 | 17.8 | 5.9 | 0 | [0 0] | 184 | 41.6 | 15.5 | 0 | [0 0] |
| Yes | 227 | 15.1 | 5.6 | -2.68*** | [-3.79 -1.58] | 225 | 33.9 | 14.6 | -7.38*** | [-10.3 -4.44] |
| Facility has adequate PPEs | | | | | | | | | | |
| No | 312 | 16.3 | 5.7 | 0 | [0 0] | 308 | 37.9 | 15.9 | 0 | [0 0] |
| Yes | 28 | 12.9 | 5.2 | -3.49** | [-5.74 -1.25] | 27 | 31.7 | 13.6 | -6.45* | [-12.6 -0.29] |
| I don't know | 74 | 17.4 | 6.2 | 1.01 | [-0.46 2.48] | 74 | 37.1 | 14.0 | -1.09 | [-5.01 2.83] |
| Facility has COVID-19 isolation ward | | | | | | | | | | |
| No | 125 | 16.5 | 6.4 | 0 | [0 0] | 124 | 37.1 | 16.4 | 0 | [0 0] |
| Yes | 275 | 16.0 | 5.5 | -0.51 | [-1.74 0.72] | 271 | 37.0 | 14.9 | 0.5 | [-2.77 3.76] |
| I don't know | 13 | 21.5 | 5.6 | 4.97** | [1.65 8.28] | 13 | 48.9 | 16.3 | 12.1** | [3.14 21.0] |
| Facility has protocol for screening for COVID-19 | | | | | | | | | | |
| No | 66 | 16.5 | 6.5 | 0 | [0 0] | 65 | 37.8 | 17.3 | 0 | [0 0] |
| Yes | 333 | 16.1 | 5.8 | -0.40 | [-1.94 1.15] | 330 | 36.9 | 15.2 | -1.75 | [-5.82 2.32] |
| I don't know | 15 | 19.3 | 2.9 | 2.82 | [-0.46 6.10] | 14 | 47.5 | 10.3 | 9.38* | [0.65 18.1] |
| Facility has protocol for managing COVID-19 | | | | | | | | | | |
| No | 147 | 16.9 | 6.2 | 0 | [0 0] | 145 | 39.2 | 17.4 | 0 | [0 0] |
| Yes | 202 | 15.9 | 5.7 | -1.00 | [-2.25 0.25] | 200 | 35.8 | 14.5 | -3.22 | [-6.51 0.073] |
| I don't know | 65 | 16.3 | 5.4 | -0.46 | [-2.17 1.24] | 64 | 38.2 | 13.3 | -0.73 | [-5.29 3.83] |
| Guidelines to report suspected COVID-19 | | | | | | | | | | |
| No | 78 | 18.3 | 6.1 | 0 | [0 0] | 76 | 43.0 | 16.3 | 0 | [0 0] |

(Continued)

**Table 2.** (Continued)

| | Stress Scores (N = 414) | | | | Burnout scores (N = 409) | | | | |
|---|---|---|---|---|---|---|---|---|---|
| | N | Mean | Sd | β | [95% CI] | N | Mean | Sd | β | [95% CI] |
| Yes | 318 | 15.8 | 5.7 | -2.42** | [-3.86 -0.98] | 315 | 36.0 | 15.0 | -6.80*** | [-10.6 -2.98] |
| I don't know | 18 | 16.0 | 6.2 | -2.26 | [-5.23 0.72] | 18 | 36.9 | 15.4 | -6.19 | [-14.1 1.77] |
| Know what to do if COVID-19 suspected | | | | | | | | | | |
| No | 21 | 18.8 | 6.1 | 0 | [0 0] | 21 | 41.3 | 14.7 | 0 | [0 0] |
| Somewhat | 118 | 18.1 | 5.4 | -0.69 | [-3.35 1.96] | 118 | 42.4 | 15.6 | 1.26 | [-5.80 8.32] |
| Yes | 275 | 15.3 | 5.8 | -3.49** | [-6.03 -0.95] | 270 | 34.9 | 14.9 | -6.25 | [-13.0 0.50] |
| Know how to manage a confirmed case of COVID-19 | | | | | | | | | | |
| No | 145 | 17.1 | 6.0 | 0 | [0 0] | 142 | 39.8 | 16.6 | 0 | [0 0] |
| Somewhat | 137 | 16.4 | 5.5 | -0.69 | [-2.05 0.67] | 137 | 37.4 | 14.8 | -2.51 | [-6.10 1.09] |
| Yes | 87 | 14.8 | 5.8 | -2.27** | [-3.82 -0.72] | 86 | 34.2 | 14.3 | -5.13* | [-9.22 -1.05] |
| Not applicable to my role | 44 | 16.3 | 6.0 | -0.80 | [-2.77 1.17] | 43 | 35.7 | 15.3 | -4.14 | [-9.32 1.04] |

95% confidence intervals in brackets

* p<0.05;

** p<0.01;

*** p<0.001

had lower burnout than those in Greater Accra and Ashanti regions. Other factors significantly associated with lower stress and burnout included appreciation, support, and communication from management; family support; confidence in being cared for if infected; training on COVID-19; availability of PPE, isolation ward, and COVID-19 guidelines; and confidence in being able to manage COVID-19 patients. Fear of infection and being female were associated with higher stress and burnout.

## Multivariate analysis

In the multivariate analysis (Tables 3 and 4), the associations between perceived preparedness with both stress and burnout were still significant. When accounting for only the demographic variables (model 1 of Table 3), providers who felt somewhat prepared and prepared had about 3- and 5-points lower stress scores respectively compared to those who did not feel at all prepared. This decreased to about 2 and 3 points, respectively, with the addition of appreciation from management and family support in model 2. In model 3, which includes fear of infection, the coefficients for somewhat prepared and prepared decreased further (β = -1.89, 95%CI:-3.49 to -0.30 and β = -2.66, 95%CI:-4.48 to -0.84) by 17% and 16% from model 2, suggesting fear of infection partially mediates the relationship between perceived preparedness and stress.

For burnout, when accounting for only the demographic variables (Table 4, model 1), providers who felt somewhat prepared and prepared had about 10 points and 14 points lower burnout scores, respectively, compared to those who did not feel at all prepared. This decreased to about 9 and 10 points, respectively, with the addition of appreciation from management and family support in model 2. In model 3, which includes fear of infection, the coefficients for somewhat prepared and prepared decreased to about 8 and 9 points (β = -7.74, 95%CI:-11.8 to -3.64 and β = -9.25, 95%CI:-14.1 to -4.41)—a 10% decrease from model 2, suggesting potential partial mediation by fear of infection. The mediated effect with the categorical preparedness variable was not significant, but it was significant with the continuous preparedness variable with the proportion of the mediated effect at 16% (Table 5).

**Table 3. Multivariable linear regression of potential predictors on perceived stress among healthcare workers in Ghana (N = 414).**

| | Perceived stress scores | | | | | |
| | Model 1 | | Model 2 | | Model 3 | |
| | β | [95% CI] | β | [95% CI] | β | [95% CI] |
|---|---|---|---|---|---|---|
| Perceived preparedness | | | | | | |
| Not at all prepared | 0.00 | [0 0] | 0.00 | [0 0] | 0.00 | [0 0] |
| Somewhat prepared | -2.95*** | [-4.59 -1.31] | -2.29** | [-3.90 -0.68] | -1.89* | [-3.49 -0.30] |
| Prepared | -4.60*** | [-6.38 -2.83] | -3.18*** | [-4.99 -1.37] | -2.66** | [-4.48 -0.84] |
| Provider type | | | | | | |
| Doctor | 0.00 | [0 0] | 0.00 | [0 0] | 0.00 | [0 0] |
| Nurse/related | 0.09 | [-1.66 1.85] | 0.04 | [-1.66 1.74] | 0.13 | [-1.54 1.80] |
| Other | 0.30 | [-1.81 2.41] | 0.23 | [-1.81 2.26] | 0.33 | [-1.69 2.35] |
| Region | | | | | | |
| Greater Accra/Ashanti | 0.00 | [0 0] | 0.00 | [0 0] | 0.00 | [0 0] |
| Northern region | -2.67** | [-4.45 -0.89] | -2.80** | [-4.54 -1.07] | -3.04*** | [-4.78 -1.30] |
| Other Northern | 0.56 | [-1.17 2.30] | 0.34 | [-1.38 2.07] | -0.08 | [-1.81 1.66] |
| other Southern | -0.75 | [-2.34 0.84] | -0.87 | [-2.41 0.66] | -1.18 | [-2.73 0.37] |
| Facility type | | | | | | |
| Teaching hospital | 0.00 | [0 0] | 0.00 | [0 0] | 0.00 | [0 0] |
| Regional/district hospital | -0.10 | [-1.76 1.56] | 0.14 | [-1.45 1.73] | 0.41 | [-1.17 2.00] |
| Health center/Other govt facility | -0.47 | [-2.41 1.47] | 0.08 | [-1.82 1.98] | 0.30 | [-1.60 2.19] |
| Private/mission facility | -0.11 | [-2.19 1.97] | 0.24 | [-1.80 2.28] | 0.26 | [-1.79 2.31] |
| Years of experience | | | | | | |
| 5 or less years | 0.00 | [0 0] | 0.00 | [0 0] | 0.00 | [0 0] |
| 6 to 10 years | 1.59* | [0.26 2.92] | 1.45* | [0.12 2.77] | 1.45* | [0.14 2.77] |
| More than 10 years | 0.35 | [-1.20 1.90] | 0.50 | [-1.04 2.04] | 0.59 | [-0.94 2.12] |
| Gender | | | | | | |
| Male | 0.00 | [0 0] | 0.00 | [0 0] | 0.00 | [0 0] |
| Female | 1.16 | [-0.040 2.35] | 1.06 | [-0.11 2.22] | 0.71 | [-0.45 1.87] |
| Marital status | | | | | | |
| Single | 0.00 | [0 0] | 0.00 | [0 0] | 0.00 | [0 0] |
| Married | -0.66 | [-1.92 0.61] | -0.78 | [-2.02 0.46] | -0.94 | [-2.17 0.28] |
| Appreciation from management | | | | | | |
| Not /somewhat appreciative | | | 0.00 | [0 0] | 0.00 | [0 0] |
| Appreciative/Very appreciative | | | -1.94** | [-3.10 -0.78] | -1.89** | [-3.05 -0.72] |
| Support from family | | | | | | |
| Not /a little supportive | | | 0.00 | [0 0] | 0.00 | [0 0] |
| Supportive/Very Supportive | | | -1.88** | [-3.13 -0.64] | -1.86** | [-3.10 -0.62] |
| Fearful of contracting COVID-19 | | | | | | |
| Not /a little fearful | | | | | 0.00 | [0 0] |
| Fearful/Very fearful | | | | | 1.89** | [0.77 3.02] |
| Constant | 19.2*** | [16.6 21.7] | 20.5*** | [17.9 23.1] | 19.5*** | [16.9 22.2] |
| Observations | 414.00 | | 414.00 | | 414.00 | |
| R-squared | 0.13 | | 0.18 | | 0.20 | |

95% confidence intervals in brackets

* p<0.05

** p<0.01

*** p<0.001

**Table 4. Multivariable linear regression of potential predictors on burnout of healthcare workers in Ghana (N = 409).**

| | Burnout scores | | | | | |
| --- | --- | --- | --- | --- | --- | --- |
| | Model 1 | | Model 2 | | Model 3 | |
| | β | [95% CI] | β | [95% CI] | β | [95% CI] |
| Perceived preparedness | | | | | | |
| Not at all prepared | 0.00 | [0 0] | 0.00 | [0 0] | 0.00 | [0 0] |
| Somewhat prepared | -10.3*** | [-14.4 -6.13] | -8.57*** | [-12.7 -4.44] | -7.74*** | [-11.8 -3.64] |
| Prepared | -14.0*** | [-18.5 -9.54] | -10.3*** | [-15.1 -5.52] | -9.25*** | [-14.1 -4.41] |
| Provider type | | | | | | |
| Doctor | 0.00 | [0 0] | 0.00 | [0 0] | 0.00 | [0 0] |
| Nurse/related | -2.07 | [-6.39 2.24] | -2.17 | [-6.26 1.93] | -1.98 | [-6.05 2.10] |
| Other | -4.00 | [-9.11 1.12] | -4.13 | [-9.05 0.80] | -3.89 | [-8.80 1.02] |
| Region | | | | | | |
| Greater Accra/Ashanti | 0.00 | [0 0] | 0.00 | [0 0] | 0.00 | [0 0] |
| Northern region | -3.29 | [-7.84 1.25] | -3.47 | [-7.96 1.02] | -4.01 | [-8.47 0.45] |
| Other Northern | 0.02 | [-5.09 5.13] | -0.53 | [-5.57 4.51] | -1.53 | [-6.49 3.43] |
| other Southern | -2.68 | [-6.97 1.61] | -2.97 | [-7.19 1.26] | -3.69 | [-7.89 0.51] |
| Facility type | | | | | | |
| Teaching hospital | 0.00 | [0 0] | 0.00 | [0 0] | 0.00 | [0 0] |
| Regional/district hospital | -1.58 | [-5.81 2.65] | -0.80 | [-4.96 3.37] | -0.20 | [-4.30 3.90] |
| Health center/Other govt facility | -1.50 | [-6.08 3.08] | 0.11 | [-4.48 4.70] | 0.59 | [-3.94 5.13] |
| Private/mission facility | -0.31 | [-5.92 5.30] | 0.79 | [-4.70 6.28] | 0.78 | [-4.68 6.24] |
| Years of experience | | | | | | |
| 5 or less years | 0.00 | [0 0] | 0.00 | [0 0] | 0.00 | [0 0] |
| 6 to 10 years | 3.25 | [-0.19 6.69] | 2.92 | [-0.49 6.32] | 2.94 | [-0.41 6.30] |
| More than 10 years | 3.35 | [-0.75 7.44] | 3.77 | [-0.24 7.78] | 4.01* | [0.027 7.98] |
| Gender | | | | | | |
| Male | 0.00 | [0 0] | 0.00 | [0 0] | 0.00 | [0 0] |
| Female | 4.29** | [1.22 7.35] | 4.07** | [1.10 7.04] | 3.28* | [0.33 6.23] |
| Marital status | | | | | | |
| Single | 0.00 | [0 0] | 0.00 | [0 0] | 0.00 | [0 0] |
| Married | -4.68** | [-8.07 -1.29] | -5.00** | [-8.34 -1.65] | -5.35** | [-8.60 -2.10] |
| Appreciation from management | | | | | | |
| Not /somewhat appreciative | | | 0.00 | [0 0] | 0.00 | [0 0] |
| Appreciative/Very appreciative | | | -5.11** | [-8.25 -1.96] | -4.95** | [-8.07 -1.83] |
| Support from family | | | | | | |
| Not/a little supportive | | | 0.00 | [0 0] | 0.00 | [0 0] |
| Supportive/Very Supportive | | | -4.99** | [-8.20 -1.77] | -4.90** | [-8.10 -1.71] |
| Fearful of contracting COVID-19 | | | | | | |
| Not/a little fearful | | | | | 0.00 | [0 0] |
| Fearful/Very fearful | | | | | 4.27** | [1.40 7.13] |
| Constant | 36.6*** | [30.1 43.2] | 40.1*** | [33.4 46.8] | 37.9*** | [31.0 44.8] |
| Observations | 409.00 | | 409.00 | | 409.00 | |
| R-squared | 0.15 | | 0.19 | | 0.21 | |

95% confidence intervals in brackets

* p<0.05

** p<0.01

*** p<0.001

**Table 5. Mediation by fear of infection among healthcare workers in Ghana.**

|  | Perceived stress score (N = 414) | | Burnout score (N = 409) | |
|---|---|---|---|---|
|  | **β** | **[95% CI]** | **β** | **[95% CI]** |
| Preparedness score |  |  |  |  |
| [1]Total effect: c | -0.12*** | [-0.18 -0.054] | -0.29*** | [-0.46 -0.12] |
| [2]Direct effect: c' | -0.097** | [-0.16 -0.032] | -0.24** | [-0.42 -0.070] |
| Mediated (Indirect) effect: c-c' | -0.020* | [-0.036 -0.0035] | -0.046* | [-0.086 -0.0065] |
| % of total effect mediated: [(c-c')/c] *100 | 17.01 |  | 15.81 |  |

[1]Includes all variables from Model 2 in Tables 3 and 4, with categorical perceived preparedness variable replaced by the continuous preparedness variables

[2]Includes all variables from Model 3 in Tables 3 and 4, with categorical perceived preparedness variable replaced by the continuous preparedness variables

Providers in the Northern region had about 3 points lower stress scores than those in Greater Accra and Ashanti regions (the COVID-19 epicenters). Females also had about 3 points higher burnout scores than males, and married providers had about 5 points lower burnout scores than unmarried providers. Perceived appreciation from management and family support were associated with about 2 points lower stress scores and about 5 points lower burnout scores, while fear of infection was associated with about 2 points higher stress scores and 4 points higher burnout scores.

## Sensitivity results

The interactions between preparedness with type of provider, appreciation from management and family support were not significant for neither stress nor burnout, suggesting the absence of conditional effects. The results obtained from using the log of burnout as the outcome, as well as that from using preparedness as a continuous variable, were consistent with the results of the untransformed burnout variable and the categorical preparedness variables respectively in their significance, direction, and magnitude of the associations. Results from the binary logistic regression based on the dichotomized stress and burnout scores were also generally consistent with the results of the continuous variables, with minor variations depending on how the variable was dichotomized (S5 Appendix). The characteristics of respondents excluded was not substantially different from those included, except on facility type where 18% of those excluded worked in a teaching hospital compared to 23% of those included (S6 Appendix).

## Discussion

We found evidence of high stress and burnout and low perceived preparedness to respond to the COVID-19 pandemic among HCWs in Ghana. Low perceived preparedness was associated with increased stress and burnout. Our findings suggest that increased fear of infection partly accounts for the effect of perceived preparedness on stress and burnout—i.e., inadequate preparation leads to fear of infection, which leads to high stress and burnout. This is, however, a small indirect effect (<20%), which is likely because other factors, including fear of poor outcomes for patients, may also be mediating the effect of preparedness on stress and burnout. In contrast, increased appreciation from management and family support decreases stress and burnout. Inadequate preparedness may, therefore, have multiplicative effects through its association with stress and burnout, which may negatively affect HCW job satisfaction, productivity, quality of care, and workforce turnover [14, 15]—outcomes that would impede Ghana's progress in containing COVID-19.

High stress and burnout among health workers in Ghana is not surprising given global evidence prior to the pandemic of provider stress and burnout—including in Ghana and other African countries [25, 47, 48]. Our prevalence of moderate (64%) and high (4%) stress among HCWs is comparable to that reported in a recent systematic review of the psychological impact of COVID-19 on HCWs and general public, which found stress to be at 40% (20%-60%) [19]. Also, compared to our findings of low (47%) and high (20%) burnout, a study of HCWs in Ghana reported burnout scores ranging from good (71.5%), alarming (12.6%), acute crisis (6.0%), and burnout (9.9%) among Accra-based HCWs; however this was prior to the COVID-19 pandemic [48]. Additionally, a study among frontline nurses caring for COVID-19 patients in Wuhan, China reported that about half of the nurses studied experienced moderate and high burnout—characterized by emotional exhaustion (60.5%) and depersonalization (42.3%) [49]. We found lower levels of moderate to emotional exhaustion (39%) and higher levels physical exhaustion (73%), although the estimates are not directly comparable given the use of different measures in the different studies.

Burnout among HCWs during COVID-19 pandemic has thus been characterized as an infection of the mind, with calls for interventions to fight the two afflictions: COVID-19 and the psychological strain experienced by medical professionals at the frontline of the response [50]. Extant studies show that factors associated with preparedness include availability of PPE, clear protocols, and isolation wards, training, and good communication from management [5, 41]. Improving these would increase perceived preparedness, decrease fear of infection, and decrease stress and burnout. Recommended steps related to preparedness include development of national and regional disaster mitigation plans to shorten the time needed to provide necessary equipment and testing; provision of adequate test kits and PPE; training on disaster management and response for HCWs; and creating a medical reserve corps of licensed individuals [50]. Such initiatives would help improve HCW preparedness to respond to COVID-19.

Similar to findings from other studies [18, 19, 51], our results suggest that feeling appreciated by management and having family support is important for HCWs' psychological wellbeing, while being unmarried, female, and working in the most impacted areas negatively affected wellbeing. Efforts are therefore needed to ensure providers feel appreciated for their role in the pandemic response and to provide additional support to HCWs who are female, unmarried, and based at the epicenters of Ghana's epidemic. Additionally, interventions are needed to increase workplace awareness of stress and burnout, self-care, availability of and access to mental health services, and to implement organizational policies and practices that prioritize HCW wellbeing [52]. In some jurisdictions, support programs such as peer-support video conferencing sessions are being offered for peer groups to discuss various issues affecting them [53]. Additionally, categorizing COVID-19 as an occupational disease, like healthcare organizations have demanded, may help improve worker protections and government accountability [26]. Interventions, like mindfulness exercises, changes to institutional culture, and workplace incentives could also improve psychological outcomes among HCWs [54]. Family support is also critical and may help lower stress and burnout [18].

## Limitations and strengths

There are some limitations to the study. The use of an online survey with a volunteer sample limits the generalizability of findings to all HCWs in Ghana. This was, however, the best option available for rapid data collection as the country was in partial lock-down due to the COVID-19 pandemic. To address this limitation, we recruited from diverse platforms such as Facebook and WhatsApp pages of different professional groups, graduation year groups, and regional

groups of HCWs. Survey links were also emailed to leaders of professional organizations and Ghana Health Service directors to share with members of their groups. Thus, our sample is diverse in terms of gender, age, years of experience, region, and facility type as shown in the sample distribution—which increases the representativeness of the findings. Moreover, our study sets the stage for future research to examine these issues in a more representative sample under circumstances that allow for probability sampling. Additionally, as with all self-reported data, social desirability and recall bias are potential limitations. The use of composite scores from validated psychosocial measures, however, helps to address this limitation. Another limitation is that this was a cross-sectional study, thus, associations described are not causal. Finally, our study only examined psychosocial outcomes; future research is needed to examine biological effects of the stress and burnout induced by COVID-19. Despite these limitations, this is the first study to our knowledge assessing perceived preparedness for COVID-19 and psychological well-being among HCWs in Africa and contributes critical findings that can help address emerging issues and challenges in the current pandemic response. It also provides a baseline for future studies in Ghana, Africa, and globally.

## Conclusions

HCWs in Ghana reported low perceived preparedness to respond to COVID-19, which was associated with increased stress and burnout. The effect of inadequate preparation on both stress and burnout is partially mediated by fear of infection. This finding is likely replicable in other low-resource settings, and potentially globally, and highlights the need for interventions to increase providers' preparedness. The government of Ghana has demonstrated commitment to addressing the needs of HCWs; however, more efforts are needed. Government and other stakeholders must institute necessary trainings, protections, and incentives to improve HCWs' psychological wellbeing and ability to respond to the pandemic. With HCW shortage in Africa, a high number of cases among these frontline workers, inadequate PPE and preparedness, and growing work demands, such interventions are critically needed to retain them and maintain the quality of care in already strained health systems. Studies in different settings examining the impact of these factors on health care quality and outcomes in the context of the pandemic are also needed. For Africa, stress and burnout have far reaching implications for the COVID-19 response. Given warnings that the continent could witness the loss of millions of lives, immediate actions are needed to strengthen health systems, train HCWs, and provide support and encouragement to boost morale.

## Supporting information

**S1 Appendix. Perceived stress items.**
(DOCX)

**S2 Appendix. Burnout items.**
(DOCX)

**S3 Appendix. Perceived preparedness items.**
(DOCX)

**S4 Appendix. Perceived knowledge items.**
(DOCX)

**S5 Appendix. Multivariable linear regression of potential predictors on stress and burnout.**
(XLSX)

**S6 Appendix. Comparing sample excluded to analytic samples.**
(DOCX)

# Acknowledgments

We wish to thank all healthcare providers who participated in the study and who helped in the survey dissemination.

# Author Contributions

**Conceptualization:** Patience A. Afulani.

**Data curation:** Patience A. Afulani, Akua O. Gyamerah, Jerry J. Nutor, Amos Laar, Raymond A. Aborigo, Hawa Malechi, John K. Awoonor-Williams.

**Formal analysis:** Patience A. Afulani.

**Funding acquisition:** Patience A. Afulani, Jerry J. Nutor.

**Methodology:** Patience A. Afulani, Jerry J. Nutor, Amos Laar, Raymond A. Aborigo, Hawa Malechi, John K. Awoonor-Williams.

**Project administration:** Patience A. Afulani, Mona Sterling.

**Supervision:** Patience A. Afulani.

**Writing – original draft:** Patience A. Afulani, Akua O. Gyamerah, Jerry J. Nutor.

**Writing – review & editing:** Patience A. Afulani, Akua O. Gyamerah, Jerry J. Nutor, Amos Laar, Raymond A. Aborigo, Hawa Malechi, Mona Sterling, John K. Awoonor-Williams.

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
