## [Decision Letter · Decision Letter 0]

8 Jan 2021

PONE-D-20-28591

Inadequate preparedness for response to COVID-19 is associated with stress and burnout among healthcare workers in Ghana

PLOS ONE

Dear Dr. Gyamerah,

Thank you for submitting your manuscript to PLOS ONE. After careful consideration, we feel that it has merit but does not fully meet PLOS ONE’s publication criteria as it currently stands. Therefore, we invite you to submit a revised version of the manuscript that addresses the points raised during the review process.

all issues raised by reviewers are required.

We look forward to receiving your revised manuscript.

Kind regards,

Vincenzo Lionetti, M.D., PhD

Academic Editor

PLOS ONE

Journal Requirements:

2. Please include captions for your Supporting Information files at the end of your manuscript, and update any in-text citations to match accordingly. Please see our Supporting Information guidelines for more information: http://journals.plos.org/plosone/s/supporting-information

Reviewers' comments:

Reviewer's Responses to Questions

**Comments to the Author**

1. Is the manuscript technically sound, and do the data support the conclusions?

Reviewer #1: Yes

Reviewer #2: No

2. Has the statistical analysis been performed appropriately and rigorously? 

Reviewer #1: Yes

Reviewer #2: No

3. Have the authors made all data underlying the findings in their manuscript fully available?

Reviewer #1: Yes

Reviewer #2: Yes

4. Is the manuscript presented in an intelligible fashion and written in standard English?

Reviewer #1: Yes

Reviewer #2: Yes

5. Review Comments to the Author

Reviewer #1: Covid-19 is the most salient topic of the year, from a health, social, and scientific point of view. The authors detailed the psychological (and psychopathological) impact of the pandemic on the most affected category in terms of mental well-being.

The statistical analysis carried out appears meticulous and well explained. The number of variables investigated is large and overall sufficient to realistically describe the distress condition of the HCWs. Through the survey, the authors characterize study participants with demographic, professional, cognitive, and emotional specifications. The constructs of stress and burnout, assessed by validated scales, undoubtedly reflect the difficulties faced by the HCWs, and the association with the insufficient perceived preparedness stimulates reflection about potential and necessary improvements in Ghana – and global – health system.

Main concerns:

I suggest adding or clarifying some information about sampling and data selection methods.

1.Since respondents have the option to skip some questions, you should specify if and how many subjects were excluded from the analysis (i.e. has a cut-off been set? Have the stress and burnout questionnaires been included only if complete?)

2.It is unclear whether the two samples (n = 414 and n = 409 for stress and burnout, respectively) overlap or if they are distinct groups (i.e did each subject answer to both questionnaires or only one of them, either burnout or stress?)

In the last line of paragraph 2.2. you wrote that "Additional study methods can be found elsewhere (41)", but in the cited article I found an identical description. In this regard, given the mention and the similarities, it would be appropriate to elucidate the possible relationship between the two manuscripts.

Minor comments:

I noticed that several times you mentioned online news sites (e.g. Reuters, GhanaWeb, CNN, Deutsche Well, ScienceDaily). I recommend, if and when possible, replacing them with references from scientific journals.

I found some typing errors in the list of references (ref. N ° 17, n ° 35). I recommend checking it out.

Reviewer #2: Evaluations:

Title

1. Does the title give clear idea about the article? No

I think it is better as “Is inadequate preparedness for, response to COVID-19 is associated with stress and burnout among healthcare workers in Ghana?”

Also, please write full words in the short title. (HCW)

Abstract

2. Does the abstract that concisely describes the content and scope of the project and identifies the project’s objective, its methodology and its findings, conclusions, or intended results? Yes

Introduction

3. Does the introduction give clear idea about the article? YES

Please write references in same fashions. Example, (line 3rd of introduction, ----people as of August 10th, 2020 (1). And line 5th of introduction ……World Health Organization (WHO)(2).

Make Times New Roman style “perceived preparedness”

Methods

4. Did methodology part is clear? NO

-I haven’t seen the importance of narrating “2.1. Context…….” under methods.

-You were collecting the data through social media. Your sampling seem non probability because participants those only your friends.

- I am afraid of the quality of data.

-Well, if sociodemographic of participants included

5. It is well if sub-headed methodology part as: study setting and period, study design, and so on.

6. Did you conduct a pretest? If yes, how and where? If no, why?

7. What language/s used for data collection?

8. You have no inclusion and exclusion criteria. Please give more clarification for this.

9. What do you think about voluntary participation in your study?

Results

10. You categorized professional type to Doctor, Nurse related and Other. First, what are others? Please specify. Second, it is not recommended to use for more than 5% for others. But yours is 17.6%

11. How you classify years of experiences and age of participants?

12. You classify marital status in to married and single. What others? (This is what I said quality of data)

13. Revise the references as the journal guideline

14. Tables titles are not written in the standard form.

6. PLOS authors have the option to publish the peer review history of their article (what does this mean?). If published, this will include your full peer review and any attached files.

Reviewer #1: No

Reviewer #2: No

---

## [Author Response · Author response to Decision Letter 0]

12 Jan 2021

Dear Dr. Lionetti,

Thank you for the opportunity to revise and resubmit our manuscript. We have given careful consideration to all the issues raised by the reviewers and revised our manuscript accordingly. Below is a point-by-point clarification and explanation of revisions made in response to the reviewers. 

Sincerely,

Manuscript authors

Response to comments

Journal comments

Response: We have revised the manuscript to meet PLOS ONE’s style requirements and name files according to the journal’s guidelines. We have also added the following sections: “contributors”, “data availability”, “Funding”, and competing interests after the conclusions section, on page 20.

2. Please include captions for your Supporting Information files at the end of your manuscript, and update any in-text citations to match accordingly. 

Response: We have renamed our Supporting Information files to reflect the journal’s guidelines, added a Supporting Information section after the references, on page 26, and revised in-text citations of the supporting information accordingly.

Reviewer #1: 

Covid-19 is the most salient topic of the year, from a health, social, and scientific point of view. The authors detailed the psychological (and psychopathological) impact of the pandemic on the most affected category in terms of mental well-being.

The statistical analysis carried out appears meticulous and well explained. The number of variables investigated is large and overall sufficient to realistically describe the distress condition of the HCWs. Through the survey, the authors characterize study participants with demographic, professional, cognitive, and emotional specifications. The constructs of stress and burnout, assessed by validated scales, undoubtedly reflect the difficulties faced by the HCWs, and the association with the insufficient perceived preparedness stimulates reflection about potential and necessary improvements in Ghana – and global – health system. 

Response: Thank you

Main concerns: 

I suggest adding or clarifying some information about sampling and data selection methods.

1.Since respondents have the option to skip some questions, you should specify if and how many subjects were excluded from the analysis (i.e. has a cut-off been set? Have the stress and burnout questionnaires been included only if complete?) 

Response: Yes, we only used data from respondents who completed all of the stress and burnout questions. 646 people started the survey (i.e answered the first question), but many did not complete the survey. For the present study, we excluded data from 232 and 237 people from the stress and burnout analytic samples. 

Many of the respondents we excluded answered only the first few questions in the survey and the stress and burnout questions were among the final set of questions. Thus, given that stress and burnout were the key outcomes in this paper, we did not think it was appropriate to impute for the missing data. We however compared the characteristics of those excluded (since the demographic questions were the initial questions and many just answered those) to those included. 

We have added Appendix 3 to our revised manuscript, which shows that the analytic sample and excluded people were not substantially different. We have described this in the methods section on page 10 and in the sensitivity analysis on page 15.

2.It is unclear whether the two samples (n = 414 and n = 409 for stress and burnout, respectively) overlap or if they are distinct groups (i.e did each subject answer to both questionnaires or only one of them, either burnout or stress?) 

Response: These two samples overlap. The difference is due to a few more missing observations on the burnout items. We have clarified this in the methods section on page 10.

-In the last line of paragraph 2.2. you wrote that "Additional study methods can be found elsewhere (41)", but in the cited article I found an identical description. In this regard, given the mention and the similarities, it would be appropriate to elucidate the possible relationship between the two manuscripts. 

Response: The first study manuscript we referenced examines factors associated with perceived preparedness of healthcare workers. We referenced it for its description of the psychometric analyses used to validate the perceived preparedness scale. In our submitted manuscript on burnout and stress, perceived preparedness is a key predictor, however we did not go into details in our methods section on the psychometric analyses/validation methods. Thus, we referenced the first manuscript for those methods. We have clarified this in the revised manuscript.

Minor comments: 

-I noticed that several times you mentioned online news sites (e.g. Reuters, GhanaWeb, CNN, Deutsche Well, ScienceDaily). I recommend, if and when possible, replacing them with references from scientific journals. 

Response: We have searched for articles in scientific journals for these references, however, we unfortunately did not find any for the Reuters, GhanaWeb, and CNN references, all of which discuss Ghanaian healthcare worker strike threats, cases, and related deaths from COVID-19. Unfortunately, these developments were reported in news articles and not in scientific reports. We have however replaced the Deutsche Welle reference with a scientific journal article (Ref # 27) and removed the Science Daily reference since it discusses a scientific study that is already referenced in our manuscript (Ref # 50).

-I found some typing errors in the list of references (ref. N ° 17, n ° 35). I recommend checking it out. 

Response: Thank you for flagging these. We have corrected the typing errors.

Thank you for your insightful comments which have strengthened our manuscript.

Reviewer #2: 

Title 

1. Does the title give clear idea about the article? No

I think it is better as “Is inadequate preparedness for, response to COVID-19 is associated with stress and burnout among healthcare workers in Ghana?” 

Response: Thank you for the suggestion. Given that titles in the form of a question are generally not advised for scientific papers that report findings, we would prefer not to frame the title in the form of a question. We chose our title based on the suggestion to use titles that are concise statements of study findings to draw the attention of readers. However, we defer to the Journal editors on whether the suggested title would be preferred for the manuscript. 

-Also, please write full words in the short title. (HCW) 

Response: We have done this.

Abstract 

2. Does the abstract that concisely describes the content and scope of the project and identifies the project’s objective, its methodology and its findings, conclusions, or intended results? Yes 

Response: Thank you

Introduction 

3. Does the introduction give clear idea about the article? YES 

Response: Thank you

-Please write references in same fashions. Example, (line 3rd of introduction, ----people as of August 10th, 2020 (1). And line 5th of introduction ……World Health Organization (WHO)(2). 

Response: We have checked the in-text references and they are consistent. The mention of “WHO” is not a reference, but rather an abbreviation for World Health Organization. We have restructured the sentence to make this clear.

Make Times New Roman style “perceived preparedness” 

Response: We have checked to confirm that all formatting is in Times New Roman style font.

Methods 

4. Did methodology part is clear? NO

-I haven’t seen the importance of narrating “2.1. Context…….” under methods. 

Response: We thought it would be appropriate to describe the context for people who might not be familiar with it to situate the study. We think this is important, but can delete it if you feel strongly about not describing the context.

-You were collecting the data through social media. Your sampling seem non probability because participants those only your friends.

Response: We acknowledge that this was a non-probability sample as described in our method. Probability sampling was not feasible given the situation at the time of the study due to the COVID-19 pandemic, but we thought it was important to conduct this study to provide data to inform policy discussions on the country’s COVID-19 response. 

It is however inaccurate to say participants are only our friends. As noted in the methods, we disseminated survey links to Facebook and WhatsApp pages of different professional groups, graduation year groups, and regional groups of HCWs, as well as to leaders of professional organizations and Ghana Health Service directors to share with members of their groups. This approach was used to ensure that we reached a diverse sample of health workers outside of our networks. The sampling method is described in our Methods section, and we discuss the limitations of a non-probability sample in our Discussion section on page 18. 

- I am afraid of the quality of data. 

Response: Per other subsequent comments, this concern appears to be related to recoding of variables. We recoded some variables to avoid very small categories for the regression analysis. We did not think it was important to go into these details. But given your concern, we have added notes to the tables, highlighting composition of recoded categories.

-Well, if sociodemographic of participants included 

Response: Sociodemographic characteristics of respondents are shown on table 1. We have modified the table title to reflect this.

5. It is well if sub-headed methodology part as: study setting and period, study design, and so on. 

Response: We have added the suggested heading(s).

6. Did you conduct a pretest? If yes, how and where? If no, why? 

Response: Yes, we pretested the survey with HCWs in Ghana before the actual study. We have added this to the methods on page 8.

7. What language/s used for data collection? 

Response: English: We have added this

8. You have no inclusion and exclusion criteria. Please give more clarification for this. 

Response: The inclusion criteria were identifying as a HCW based in Ghana, which by default meant exclusion criteria was not identifying as a HCW, and not based in Ghana. We have noted the eligibility criteria in the methods to clarify this on page 8.

9. What do you think about voluntary participation in your study? 

Response: As noted in our limitations, voluntary participation is a limitation as volunteers may respond differently from non-volunteers. But as mandated by ethical guidelines from the Institutional Review Board, participation in any study should be voluntary and so this does not invalidate the findings.

Results 

10. You categorized professional type to Doctor, Nurse related and Other. First, what are others? Please specify. Second, it is not recommended to use for more than 5% for others. But yours is 17.6% 

Response: We acknowledge this concern. As noted in text, the other professionals, included medical laboratory professionals, disease control officers, nutritionists and other allied health care workers. We have also added this as footnotes in the tables. We grouped these together, because the number of people in these sub-groups was small compared to doctors and nurses and separating them out resulted in very small categories.

11. How you classify years of experiences and age of participants? 

Response: The classification was based both on the distribution of the data and to obtain conceptually meaningful categories. We however note in the text that “The average age of respondents was 34.2 years (SD=6.0), with 8.2 years of professional experience (SD=5.6).”

12. You classify marital status in to married and single. What others? (This is what I said quality of data) 

Response: There were too few respondents in the other groups, so these were recoded into the two categories. The married category includes 10 people (2%) who were previously married (widowed, separated, or divorced). We have added this as footnotes in the table.

13. Revise the references as the journal guideline 

Response: The Journal’s reference guideline is Vancouver, which we used to format our references. We have reviewed the references and verified that they meet the Vancouver guidelines.

14. Tables titles are not written in the standard form. 

Response: We have edited the Table titles to fit a more standard form. Thank you for your insightful comments to strengthen the paper.

---

## [Decision Letter · Decision Letter 1]

28 Mar 2021

PONE-D-20-28591R1

Inadequate preparedness for response to COVID-19 is associated with stress and burnout among healthcare workers in Ghana

PLOS ONE

Dear Dr. Gyamerah,

Thank you for submitting your manuscript to PLOS ONE. After careful consideration, we feel that it has merit but does not fully meet PLOS ONE’s publication criteria as it currently stands. Therefore, we invite you to submit a revised version of the manuscript that addresses the points raised during the review process.

ACADEMIC EDITOR: Issue related to discussion is required.

We look forward to receiving your revised manuscript.

Kind regards,

Vincenzo Lionetti, M.D., PhD

Academic Editor

PLOS ONE

Journal Requirements:

Additional Editor Comments (if provided):

The authors have improved the manuscript, but in light of revised version it is important that authors highlight potential comorbidities that affect the enrolled patients (i.e.: cardiovascular disease, obesity). Indeed, recent preclinical study demonstrated that psychosocial stress exerts bigger detrimental effect in the presence of overweight/obesity. Moreover, cardiac function is also impaired following exposure to psychosocial stress since significant oxidative stress occurs in both brain and heart (please see EBioMedicine. 2019 Sep;47:384-401). Since COVID19 induce severe inflammatory response and impair cardiovascular function, authors should discuss their data in light of abovementioned study.

Reviewers' comments:

Reviewer's Responses to Questions

**Comments to the Author**

1. If the authors have adequately addressed your comments raised in a previous round of review and you feel that this manuscript is now acceptable for publication, you may indicate that here to bypass the “Comments to the Author” section, enter your conflict of interest statement in the “Confidential to Editor” section, and submit your "Accept" recommendation.

Reviewer #1: All comments have been addressed

2. Is the manuscript technically sound, and do the data support the conclusions?

Reviewer #1: Yes

3. Has the statistical analysis been performed appropriately and rigorously? 

Reviewer #1: Yes

4. Have the authors made all data underlying the findings in their manuscript fully available?

Reviewer #1: Yes

5. Is the manuscript presented in an intelligible fashion and written in standard English?

Reviewer #1: Yes

6. Review Comments to the Author

Reviewer #1: I was pleased to receive and review for the second time the article entitled "Inadequate preparedness for response to COVID-19 is associated with stress and burnout among healthcare workers in Ghana", regarding psychopathological impact of the pandemic on the most affected category in terms of mental well-being.

The authors have exhaustively clarified my doubts, responding privately to my comments or adding specific requests in the text. In particular: information regarding sampling methods, exclusion / inclusion criteria for participants, composition of questionnaires, number and characteristics of drop outs, "overlapping" of the samples for stress and burnout, relationship with the previous paper, references from news sites .

I believe that the information added corresponds to my requests and is sufficient to strengthen the manuscript.

7. PLOS authors have the option to publish the peer review history of their article (what does this mean?). If published, this will include your full peer review and any attached files.

Reviewer #1: No

---

## [Author Response · Author response to Decision Letter 1]

30 Mar 2021

Dear Dr. Lionetti,

Thank you for the opportunity to revise and resubmit our manuscript. We have given consideration to the citation suggestion you made and have revised our manuscript accordingly. Below is a clarification and explanation of the revision made in response to the review. 

Sincerely,

Manuscript authors

Response to comments

Additional Editor Comments (if provided):

The authors have improved the manuscript, but in light of revised version it is important that authors highlight potential comorbidities that affect the enrolled patients (i.e.: cardiovascular disease, obesity). Indeed, recent preclinical study demonstrated that psychosocial stress exerts bigger detrimental effect in the presence of overweight/obesity. Moreover, cardiac function is also impaired following exposure to psychosocial stress since significant oxidative stress occurs in both brain and heart (please see EBioMedicine. 2019 Sep;47:384-401). Since COVID19 induce severe inflammatory response and impair cardiovascular function, authors should discuss their data in light of abovementioned study.

Response: We acknowledge this concern. But our study did not enroll patients. Rather it enrolled healthcare workers to examine how their preparedness to respond to COVID-19 is associated with their stress and burnout using psychosocial measures. We have carefully reviewed the paper referenced (EBioMedicine. 2019 Sep;47:384-401) and it does not appear directly relevant to our paper given its focus on biological mechanisms and effects of psychosocial stress on obese mice. Thus, citing the suggested paper would appear out of context in our paper, which focuses on how preparedness level of healthcare workers is associated with psychosocial measures of burnout and stress, not the effects of stress and burnout. We have, however, acknowledged in the limitations section that exploring the effects of stress and burnout experienced by healthcare workers and the associated biological mechanisms was beyond the scope of our paper and that future research may be needed to examine these mechanisms.

Reviewer 1 Comments to the Author

1. If the authors have adequately addressed your comments raised in a previous round of review and you feel that this manuscript is now acceptable for publication, you may indicate that here to bypass the “Comments to the Author” section, enter your conflict of interest statement in the “Confidential to Editor” section, and submit your "Accept" recommendation.

Reviewer #1: All comments have been addressed

Response: Thank you

2. Is the manuscript technically sound, and do the data support the conclusions?

Reviewer #1: Yes

Response: Thank you

3. Has the statistical analysis been performed appropriately and rigorously?

Reviewer #1: Yes

Response: Thank you

4. Have the authors made all data underlying the findings in their manuscript fully available?

Reviewer #1: Yes

Response: Thank you

5. Is the manuscript presented in an intelligible fashion and written in standard English?

Reviewer #1: Yes

Response: Thank you

6. Review Comments to the Author

Reviewer #1: I was pleased to receive and review for the second time the article entitled "Inadequate preparedness for response to COVID-19 is associated with stress and burnout among healthcare workers in Ghana", regarding psychopathological impact of the pandemic on the most affected category in terms of mental well-being.

The authors have exhaustively clarified my doubts, responding privately to my comments or adding specific requests in the text. In particular: information regarding sampling methods, exclusion / inclusion criteria for participants, composition of questionnaires, number and characteristics of drop outs, "overlapping" of the samples for stress and burnout, relationship with the previous paper, references from news sites .

I believe that the information added corresponds to my requests and is sufficient to strengthen the manuscript.

Response: Thank you.

7. PLOS authors have the option to publish the peer review history of their article (what does this mean?). If published, this will include your full peer review and any attached files.

Do you want your identity to be public for this peer review? For information about this choice, including consent withdrawal, please see our Privacy Policy.

Reviewer #1: No

---

## [Editor Report · Decision Letter 2]

5 Apr 2021

Inadequate preparedness for response to COVID-19 is associated with stress and burnout among healthcare workers in Ghana

PONE-D-20-28591R2

Dear Dr. Gyamerah,

We’re pleased to inform you that your manuscript has been judged scientifically suitable for publication and will be formally accepted for publication once it meets all outstanding technical requirements.

Kind regards,

Vincenzo Lionetti, M.D., PhD

Academic Editor

PLOS ONE
---

## [Editor Report · Acceptance letter]

8 Apr 2021

PONE-D-20-28591R2 

Inadequate preparedness for response to COVID-19 is associated with stress and burnout among healthcare workers in Ghana 

Dear Dr. Gyamerah:

I'm pleased to inform you that your manuscript has been deemed suitable for publication in PLOS ONE. Congratulations! Your manuscript is now with our production department. 

Kind regards, 

on behalf of

Prof. Vincenzo Lionetti 

Academic Editor

PLOS ONE